# Effects of Atrial Fibrillation Radiofrequency Ablation in Patients Aged > 75 Years Undergoing Mitral Valve Surgery

**DOI:** 10.3390/jcm12051812

**Published:** 2023-02-24

**Authors:** Carlo Rostagno, Camilla Tozzetti, Enrico Carone, Pierluigi Stefàno

**Affiliations:** 1Dipartimento Medicina Sperimentale e Clinica, Università Firenze, 50134 Firenze, Italy; 2Cardiochirurgia AOU Careggi Firenze, 50134 Firenze, France

**Keywords:** mitral valve surgery, atrial fibrillation, radiofrequency ablation, heart failure, rheumatic valve disease, survival

## Abstract

Background: Few data exist about the efficacy of radiofrequency (RF) maze procedures in elderly patients with atrial fibrillation (AF) undergoing surgery for mitral valve disease. The aim of the present investigation was to evaluate the effects of AF ablation associated with mitral valve surgery on the recovery and long-term maintenance of sinus rhythm in elderly patients aged > 75 years. Moreover, we evaluated the effects on survival. Methods and results: This study included 96 consecutive patients with AF (42 men and 56 women) aged > 75 years (mean age 78 ± 3) who underwent RF ablation associated with mitral valve surgery (group I). This group was compared to 209 younger patients (mean age 65 ± 8 years) treated in the same period (group II). Baseline clinical and echocardiographic characteristics were similar in the two groups. Four patients died during hospitalization, one aged > 75 years. In surviving patients at the end of the follow-up period, sinus rhythm was present respectively in 64% of the elderly and 74% of younger patients (*p* = 0.778). The rate of persistence of sinus rhythm without AF recurrences (38% vs. 41%, *p* = 0.705) was similar in the two groups. After surgery, sinus rhythm was frequently never regained in aged patients (27% vs. 20%, *p* = 0.231). Elderly patients more frequently needed permanent pacing and had more hospitalizations and a higher number of non-AF atrial tachyarrhythmias. At eight-year follow-up, survival was lower in older patients (48% aged > 75 vs. 79% aged < 75 years). Conclusion: Elderly patients had a similar long-term rate of stable sinus rhythm maintenance in comparison to younger patients after AF radiofrequency ablation associated with mitral valve surgery. However, they needed more frequent permanent pacing and had higher rates of hospitalizations and post-procedural atrial tachyarrhythmias. The effects of survival are difficult to evaluate due to the different life expectancies of the two groups.

## 1. Introduction

In the last decade, due to population aging, the need for heart surgery in elderly patients (>75 years) has increased significantly [1]. A higher prevalence of AF in elderly people is a long-term recognized phenomenon [2]. After mitral valve surgery without ablation, spontaneous rhythm restoration occurs in no more than 20% of patients. The persistence of atrial fibrillation (AF) is associated with decreased functional capacity, an increased risk of embolization [3,4,5], and higher mortality [6,7]. Radio-frequency ablation has been consistently demonstrated to restore sinus rhythm in patients undergoing mitral valve surgery [8,9,10]. The long-term success rate is influenced by several variables, for example, completeness of line ablation, left atrium dimensions, concomitant surgery other than mitral valve repair/replacement, and finally, rheumatic etiology. Moreover, different results were reported in patients treated with mono- or bipolar techniques [11]. Little information is available about the effects of age since almost all studies included patients aged < 70 years. Catheter ablation of atrial fibrillation has been safely and successfully performed in elderly patients with and without underlying heart valve disease who do not need surgical treatment with results similar to younger patients.

The aim of the present investigation was to prospectively assess the effects of monopolar-bipolar radiofrequency ablation of AF performed during mitral valve surgery in patients aged > 75 years compared with a younger control group. At an average 8-year follow-up, the rate of persistence of sinus rhythm and the frequency of clinically documented recurrences of AF were compared between the groups. We also compared the need for further hospitalization due to cardiac events and overall survival. Finally, in surviving patients. we examined the relationship between the persistence of sinus rhythm and functional capacity.

## 2. Materials and Methods

### 2.1. Patient Population

Between January 2010 and December 2015, monopolar or bipolar radiofrequency ablation associated with mitral valve surgery was performed at the Heart Surgery Department of the Azienda Ospedaliera Universitaria di Careggi (AOU) in 301 patients with AF. Informed consent was obtained before they participated in the study. The study was conducted in accordance with the Declaration of Helsinki, and the protocol was approved by the Ethics Committee of AOU Careggi.

Functional capacity was expressed as NHYA functional class. All patients underwent transthoracic echocardiography (Sequoia C256 Accuson Siemens, Mountain View, CA, USA). In each patient, the following dimensions were measured: left atrium AP diameter (mm), 2D left and right atrium area (cm^2^), and left ventricular ejection fraction (LVEF). Since most patients were in AF, we considered the average value of five measurements. End-diastolic and end-systolic images were synchronized on ECG. Pulmonary systolic pressure (PAP) was calculated by adding the RV/RA pressure gradient to the estimated right atrial pressure assessed by inferior vena cava diameter and response to respiratory acts.

The follow-up of this prospective study was conducted as outpatients with clinical, electrocardiographic, and echocardiographic examinations at one and six months and thereafter yearly. The overall duration of follow-up was 8 years.

### 2.2. Radiofrequency Ablation Procedure

Medtronic Cardioablate surgical ablation systems (Medtronic, Minneapolis, MN, USA) were used for monopolar and bipolar treatment. Access to the inside of the left atrium was gained through a standard atriotomy. After left atrial appendage (LAA) excision, ablation lines were performed. A detailed description of left-sided ablation lines has been previously reported [12]. The amount of cardiopulmonary bypass time required for ablation was, on average, 15 ± 7 min.

### 2.3. Postoperative Management

Standard antiarrhythmic prophylaxis consisted of i.v. and thereafter orally administered amiodarone according to a previously reported protocol [12]. Patients with persistent AF despite optimal medical therapy before discharge underwent at least one attempt of external cardioversion with biphasic DC shock. Oral anticoagulation was given to maintain the international normalized ratio between 2.5 and 3.5 for the first 6 months in all patients and for life in patients who received mechanical valves or who had AF persistence, or both.

### 2.4. Follow-Up

Follow-up visits were performed at 3, 6, and 12 months after surgery and annually thereafter. Between visits, their referring physician followed patients on a regular basis, and routine ECGs were obtained at each clinic visit regardless of symptoms. Between visits, all patients were encouraged to seek 12-lead ECG documentation for any symptom suggestive of AF/atrial flutter recurrence, and a physician routinely performed trans-telephonic monitoring of any symptoms and complications.

The follow-up evaluation consisted of a detailed history, physical examination, and 24-h Holter monitoring. Success and AF recurrence were defined following the HRS/EHRA/ECAS expert consensus document [13].

### 2.5. Statistical Analysis

Continuous variables were presented as means ± SD, while categorical variables were reported as percentages. Continuous variables were compared with Student’s 2-tailed unpaired samples *t*-test. Categorical variables were compared using the chi-squared test or Fisher’s exact test if appropriate. Kaplan–Meier curves were used for the survival analysis. Differences between groups were compared using the log-rank test.

A probability value <0.05 was considered significant. Statistical analyses were performed with SPSS 22.0 software (SPSS, Inc., Chicago, IL, USA).

## 3. Results

The study included 95 patients (54 men and 42 women, mean age 78 ± 3 years). This group was compared to 206 younger patients (111 males and 95 females, mean age 65 ± 8 years). The characteristics of the two groups are reported in Table 1. The duration of atrial fibrillation was significantly longer in younger patients (54 vs. 26 months). Pulmonary pressure was slightly higher in elderly patients, while at the time of surgery, the degree of functional impairment (most patients in the III-IV NHYA class) did not differ between the two groups.

The etiology of mitral valve disease is reported in Table 2. No significant differences were found between younger and elderly patients. Rheumatic disease was still the prevalent indication for mitral valve surgery in both groups, though valve prolapse accounted for about a quarter of patients in elderly and younger patients.

Surgical techniques are reported in Table 3. Isolated procedures on the mitral valve were performed in 43% of elderly patients in comparison to 53% of the control group. Mitral valve replacement was more frequently performed in younger patients, while tricuspid valve repair for severe tricuspid regurgitation was more frequently performed in elderly patients (most, >85%, were performed according to the Kay technique). Mitral regurgitation secondary to coronary heart disease was present in 17.3% of the elderly vs. 10% of the control group. Radiofrequency ablation was performed using a unipolar probe in 56% of cases, and bipolar ablation was performed in the remaining 44% of cases. The proportion of patients undergoing monopolar vs. bipolar ablation did not differ between elderly and younger patients. Furthermore, the clinical characteristics of patients undergoing the two techniques were similar between the subgroups.

### 3.1. Rhythm Analysis

A total of 60/92 subjects (65%) aged > 75 years were in sinus rhythm at hospital discharge in comparison to 143/209 (68%) of younger patients (*p* = 0.360). In surviving patients at the end of the follow-up period, sinus rhythm was present in 64% of elderly and 74% of younger patients (*p* = 0.778). Sinus rhythm never recovered after ablation and electrical CV attempts in 27 (28%) individuals aged > 75 years and in 45 (21%) younger patients (*p* = 0.231). The recurrence rate was 32% and 35% in the two groups (*p* = 0.705) (Table 4). In both groups, the persistence of stable sinus rhythm was less frequent in patients with rheumatic valve disease. Additionally, these patients had a higher rate of atrial fibrillation recurrence. We did not find any significant difference in long-term results in patients treated with monopolar or bipolar ablation.

In elderly patients, atrial tachyarrhythmias different from atrial fibrillation were more frequent (11.5% vs. 4%, *p* = 0.025) than in younger patients. Permanent pacing was also more frequently needed in the older group (22 vs. 11%, *p* = 0.014) (Table 5).

Ischemic stroke occurred in six patients (2%) during the follow-up period, with 3 in each group. In patients with permanent AF, at the moment of the stroke, 3 had INR values below the therapeutic range. One of the two patients in SR had severe carotid stenosis. A higher rate of hospitalization due to cardiac causes during the follow-up period was found in the elderly group (41 vs. 30%, *p* = 0.004).

### 3.2. Functional Capacity

The baseline mean NYHA class was 3 in both groups before surgery and ablation. A significant improvement in functional capacity was found in patients in sinus rhythm at the end of follow-up (both patients with stable sinus rhythm and with AF recurrences) but not in patients who never recovered sinus rhythm (mean NYHA class was, respectively, 1.3 ± 0.4 vs. 2.3 ± 0.6, *p* < 0.001)

### 3.3. Mortality

At eight years of follow-up, overall survival was 78% (Figure 1). Eighty-eight patients died: in 62%, death was due to cardiac causes, and in the other 38%, the cause of death was not cardiac or unknown. Mortality was close to 50% in patients aged > 75 years in comparison to 20% in younger patients. Survival curves, however, began to diverge only after the first 1500 days of follow-up, a phenomenon related to the decreased life expectancy of elderly patients. Mortality in patients with rheumatic disease was higher than in those suffering from mitral valve prolapse (22% vs. 8%, *p* = 0.01). Tricuspid valve repair, more frequently performed in aged patients with pulmonary hypertension, was associated with significantly higher mortality. Preoperative NHYA class was not related to survival both in elderly and younger patients, while failure to restore sinus rhythm with RF ablation was related to a worse prognosis independently of age. Mortality was 44% in patients in AF who never recovered SR in comparison to 16% of those in stable sinus rhythm after discharge. There was no difference in survival rate between patients with stable sinus rhythm during follow-up and patients with AF recurrences. Finally, no differences in mortality were observed between patients treated with monopolar or bipolar ablation.

## 4. Discussion

The increasing number of aged patients who need heart surgery and suffer from atrial fibrillation raises the question of the cost-effectiveness of ablative procedures in elderly patients [13]. A recent study demonstrated that radiofrequency ablation in patients with heart failure and atrial fibrillation significantly decreased the combined endpoint of death and hospitalization for worsening heart failure [14]. Death due to cardiovascular causes was two-fold higher in patients treated with medical therapy in comparison to patients who underwent ablation. Non-treated atrial fibrillation in patients undergoing valve surgery is associated with a higher risk of stroke and mortality, and age has been demonstrated as a relevant independent factor for poorer outcomes [3,13]

Several studies, including matched-controlled and randomized trials [15], have consistently demonstrated that radiofrequency ablation associated with mitral valve surgery, but also with other surgical procedures, maintains sinus rhythm at short- and long-term follow-up [16,17], decreases the risk of stroke, and, with a lower degree of evidence, improves long-term survival. The rate of clinical success, however, is significantly influenced by the population selected [18,19]. Female gender, duration of atrial fibrillation above 24 months, increased left atrial dimensions (LA M-mode diameter > 54 mm, left atrial area > 24 cm^2^), rheumatic valve disease, and NYHA class were associated with a higher rate of ablation failure [20].

Analysis of the wide range of literature published in the last 15 years, however, shows that in most published studies, the mean age of included patients was between 55 and 65 years; therefore, results are not applicable to the elderly population.

Extensive evidence exists at present indicating that catheter ablation of atrial fibrillation can be safely and successfully performed in the elderly with and without underlying heart valve disease who do not need surgical treatment. Wang et al. [21], in a propensity score study, matched 347 pairs of patients aged > 75 years undergoing or not undergoing ablation. Ablation was associated with a lower risk of a composite outcome of all-cause death, non-fatal stroke, and peripheral embolism (hazard ratio (HR) = 0.40; 95% confidence interval (CI): 0.19–0.85), all-cause death (HR = 0.13 95% CI: 0.04–0.43), and major bleeding (HR = 0.23; 95% CI: 0.12–0.67). Nademanee et al. [22] evaluated 587 elderly patients (age ≥ 75 years) with AF. Three hundred and twenty-four were eligible for ablation. The 261 (group 1) who underwent ablation were compared with the other 63 patients (group 2) who declined or were not suitable for ablation. Normal sinus rhythm, stroke, death, and major bleeding were the main endpoints. At a mean follow-up of 3 ± 2.5 years 216 (83%) of group 1 patients remained in sinus rhythm in comparison to 14 of group 2 patients (22%, *p* < 0.001). At five years, survival was 87% in patients in sinus rhythm, 52% in patients with AF, and finally, 42% in patients who did not undergo ablation. Overall, the efficacy rates of catheter ablation in restoring sinus rhythm were reported to be between 75 and 85% [23,24,25]. High rates of atrial fibrillation control in the elderly were obtained despite the higher prevalence of structural heart disease and the higher prevalence of persistent atrial fibrillation. The efficacy endpoints, however, were not uniform, and in many studies, effective rhythm control required the continued use of antiarrhythmics post-ablation [23,24,25]. Nevertheless, results of catheter ablation in elderly patients suggest that age is not a contraindication to treatment.

Less is known about ablation associated with heart and, in particular, mitral valve surgery. Ablation is performed in less than 60% of patients with AF undergoing mitral valve surgery, and multivariate regression has demonstrated that age and comorbidities are strong predictors of a lower probability of performing concomitant AF ablation [26]. In their study, Petersen et al. [27] reported that freedom from atrial fibrillation at 12 months after surgery associated with AF ablation was between 62% and 72%. This was independent of age except for elderly patients undergoing concomitant coronary artery bypass grafting surgery. Double-valve procedures (odds ratio, 3.48; *p* = 0.020), preoperative persistent atrial fibrillation (odds ratio, 2.43; *p* = 0.001), and coronary artery bypass grafting surgery in elderly patients (odds ratio, 2.03; *p* = 0.009) were risk factors for the recurrence of atrial fibrillation. Lin et al. [28] evaluated the effects of bipolar radiofrequency ablation in patients aged > 65 years undergoing mitral valve replacement according to frailty status. Even if freedom from AF after 1 year was not different in the frail group compared to the non-frail group (75.1% vs. 73.5%), the frail group had a higher adjusted risk for all-cause mortality and all-cause hospitalization. Rates of cardiovascular death, stroke or non-CNS embolism, and cardiovascular hospitalization were similar between the two groups.

The study by McGregor et al. [29] compared elderly patients (mean age 78.5 ± 2.8 years) undergoing MV repair or replacement with a younger group. In elderly patients, MV replacement was more frequent than repair; additionally, elderly patients more frequently underwent other surgical procedures concomitantly. Baseline clinical conditions were more compromised in the elderly group (lower BMI, higher rates of hypertension, previous myocardial infarction, and heart failure). Major complications after surgery and 30-day mortality were more frequent in the elderly (23% vs. 14%, *p* = −0.017 and 6% vs. 2%, *p* = 0.026, respectively). Freedom from atrial fibrillation and antiarrhythmic drugs (AADs) was lower in elderly patients at 4 years (65% vs. 79%, *p* = 0.043).

Results from the present prospective study suggest that at the end of an eight-year follow-up period, the number of patients in stable sinus rhythm was non-significantly different between the two groups. Furthermore, the number of patients with AF recurrences and those who never regained sinus rhythm after surgery were similar in patients aged > 75 years and in the younger group. These data suggest that advanced age is not to be considered a contraindication to RF ablation during surgery for mitral valve disease; moreover, the results on rhythm control during follow-up are not different from those observed in younger patients. All patients had left atrial auriculectomy, and this may explain the small number of embolic complications (six, three for each group) found in the present study. Among patients in AF, 3/4 had INR values below the therapeutic range, while a critical stenosis of the internal carotid artery was found in one of the two patients in sinus rhythm.

Elderly patients had a two-fold higher need for permanent pacing than younger patients. Additionally, they needed more frequent hospitalizations and had a higher number of atrial tachyarrhythmias other than atrial fibrillation. The need for permanent pacing in patients aged > 75 years is consistent with previous data both after surgical [30,31] and catheter ablation of atrial fibrillation. In the study by De Rose et al., 14.4% of patients received a PPM within the first year after ablation associated with mitral valve surgery. A pacemaker was implanted in 7.7% of patients randomized to mitral valve surgery alone, 16.1% of patients who received mitral valve surgery + pulmonary vein isolation, and 25% of patients who received mitral valve surgery + a bi-atrial maze. Ablation, multivalve surgery, and New York Heart Association functional (NYHA) functional class III/IV were independent risk factors for PPM implantation. The need for PPM was associated with a higher risk of 1-year mortality (HR: 3.21; 95% CI: 1.01 to 10.17; *p* = 0.05) after adjustment for randomization assignment, age, and NYHA functional class. Higher rates of hospitalization and atrial arrhythmias other than atrial fibrillation were more frequently reported in elderly patients. More advanced atrial myopathy in older individuals may lead to a more complex substrate, which may affect the probability of the onset of atrial arrhythmias other than atrial fibrillation.

More difficult to evaluate is whether AF ablation may improve survival in elderly patients. Several studies have suggested increased survival in elderly patients after AF ablation in comparison to valve surgery alone [31]. Additionally, the persistence of sinus rhythm after surgery was associated with an improvement in functional capacity and increased survival in comparison to patients who remained in permanent AF. In a previous investigation from our group, mortality at five years was 30% in patients with permanent AF; otherwise, survival was not significantly different (close to 90%) between patients with stable SR and those in whom SR was present at the last follow-up visit, despite clinical documentation of at least a recurrence of AF requiring cardioversion [32]. Results from the present study show that survival curves between the two groups do not diverge within the first 1500 days after surgery. Most previous papers dealing both with catheter and surgical treatment had a limited follow-up duration (no more than 4 years), which was half of the length considered in this study; therefore, age-related increases in mortality may have been overlooked. In our opinion, the increase in late mortality in elderly patients may be mainly related to the physiological decrease in life expectancy rather than to the effects of surgery and AF ablation.

### Limitations

The main limitation of this prospective study is that it was conducted in a single high-volume center, and patients did not undergo randomization before surgery; therefore, both elderly and younger patients lack a control group. Therefore, we did not exclude possible confounding variables that may influence the results of the ablation procedure and the determination of clinical outcomes. It is impossible to know how many of the patients would convert to sinus rhythm spontaneously, and, more relevantly, it is impossible to know the effects of confounding variables (e.g., the different degrees of hemodynamic impairment). For example, tricuspid repair is associated with a lower rate of sinus rhythm restoration and with increased long-term mortality. In patients with severe pulmonary hypertension and right ventricular dysfunction, indications for tricuspid valve repair should be accurately evaluated. The overall limited number of patients undergoing isolated mitral valve repair or replacement does not allow us to assess whether a technical approach to mitral valve surgery may affect the results of the ablation procedure. Finally, although we did not find any significant difference between the results obtained with monopolar and bipolar ablation, again, the absence of randomization does not allow us to draw definite conclusions about the different results of radio-frequency ablation techniques.

## 5. Conclusions

Results from the present investigation support the efficacy of RF ablation (mono and bipolar) in restoring SR in patients with AF undergoing mitral valve surgery. We did not find significant differences between patients aged > 75 years and younger patients in terms of maintenance of SR and the recurrence rate of atrial fibrillation. Embolic complications are negligible in both groups. Aged patients have a two-fold risk of need for definitive pacing and a higher rate of atrial tachyarrhythmias, as well as a higher rehospitalization rate. The length of follow-up does not allow us to draw conclusions on the effects of the procedure on survival since the divergence of survival curves after the first 1500 days of follow-up may be related only to the decreased life expectancy of elderly patients.

## Figures and Tables

**Figure 1 jcm-12-01812-f001:**
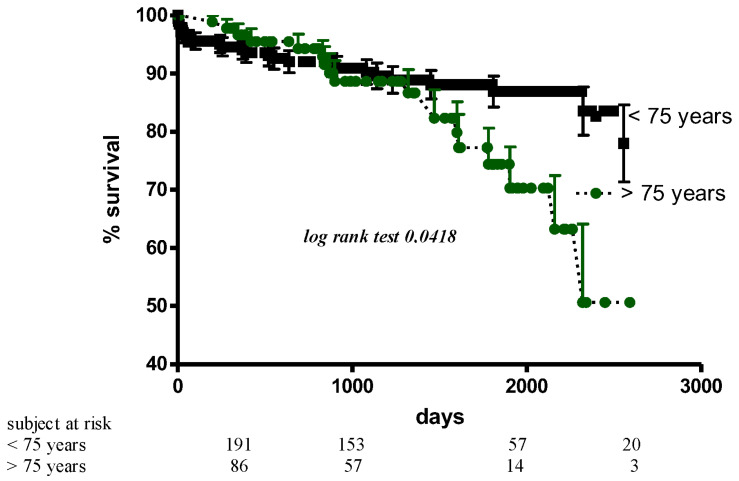
Kaplan–Meier survival curves according to age.

**Table 1 jcm-12-01812-t001:** Clinical characteristics of patients.

	>75 Years *n* = 96	<75 Years *n* = 209	
Age (years)	78 ± 3	65 ± 8	<0.001
Gender (male/female)	54/42 (56%)	111/95 (53%)	0.774
Persistent AF/paroxysmal AF	81/15 (83%)	183/26 (87%)	0.475
AF duration before surgery (months)	26 ± 28	54 ± 34	<0.001
Left ventricular EF (%)	52 ± 7	53 ± 9	0.334
Left atrium diameter (mm)	47 ± 7	46.5 ± 7	0.383
Left atrium area (cm^2^)	31.0 ± 7	32.7 ± 10.6	0.156
Right atrium area (cm^2^)	21.4 ± 5	21.2 ± 7	0.854
Systolic pulmonary pressure (mmHg)	47 ± 15	43 ± 14	0.035
NHYA class	3.1 ± 0.7	3 ± 0.55	0.177

AF = atrial fibrillation, EF = ejection fraction, NYHA = New York Heat Association.

**Table 2 jcm-12-01812-t002:** Etiology of mitral valve disease.

	>75 Years *n* = 96	<75 Years *n* = 209	*p*
Mitral valve prolapse	22 (23%)	51 (24%)	0.885
Rheumatic mitral valve disease	25 (26%)	64 (31%)	0.494
Mitro-aortic rheumatic valve disease	18 (19%)	55 (26%)	0.193
Ischemic mitral regurgitation	17 (17.2%)	21 (10%)	0.072
Mitral regurgitation associated with DCM	13 (14%)	14 (7%)	0.063
Other (including tricuspid valve repair)	1 (0.8%)	4 (2%)	0.823

DCM—dilated cardiomyopathy.

**Table 3 jcm-12-01812-t003:** Intervention performed.

Intervention	>75 Years *n* = 96	<75 Years *n* = 209	*p*
Mechanical mitral valve replacement	8 (9%)	39 (19%)	0.025
Mitral valve repair	32 (34%)	71 (34%)	0.910
Mitral and aortic valve replacement	15 (16%)	42 (20%)	0.422
Mitral valve repair and CABG	20 (21%)	31 (15%)	0.196
Mitral valve replacement and CABG	4 (2%)	9 (5%)	0.445
Other (including tricuspid valve repair)	17 (18%)	14 (7%)	0.007

CABG = Coronary artery by-pass grafting.

**Table 4 jcm-12-01812-t004:** Long-term results of AF ablation.

	Stable Sinus Rhythm	AF Recurrences	Never Recovered Sinus Rhythm	*p*
Aged < 75 years	86	75	45	0.822
Aged > 75 years	37	31	27	

**Table 5 jcm-12-01812-t005:** Complications.

Complications	>75 Years	<75 Years	*p*
Definitive pacemaker	21 (22%)	23 (11%)	0.014
Hospitalizations	45 (41%)	62 (30%)	0.004
Other atrial arrhythmias	11 (11.5%)	9 (4%)	0.025
Stroke	3 (3.1%)	3 (1.4%)	0.38

## Data Availability

Data may be available on request.

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
