# Peer review of "Effects of Atrial Fibrillation Radiofrequency Ablation in Patients Aged > 75 Years Undergoing Mitral Valve Surgery"

_jcm, 2023, doi:10.3390/jcm12051812_

Round 1

Reviewer 1 Report

The topic is interesting given the prevalnce of AF according to age and after cardiac surgery. The main message is that Maze ablation in elderly patients seems safe and efficacious. However, the results can be biased by difference in MV repair vs replacement, and this should be analyzed or acknowledged.

Minor considerations:

- it is not clear wether the study is prospective or restrospective.

- please add % in table 1

- some spelling errors

Author Response

The difference between valve replacement and repair was not statistically significant between groups. Within the same group again no difference were found between repair and replacement

The study is a prospective investigation and this is reported in the final part of introduction.  It is added in the method section 

Per cent was added tom table I when appropriate

Text was revised 

Reviewer 2 Report

The authors well describe the efficacy of RF maze procedure in elderly population.  They included 96 consecutive patients with AF (42 men  and 56 women) aged > 75 years (mean age 78+3) who underwent RF ablation associated with mitral 13 valve surgery (group I). This group was compared to 209 younger patients (mean age 65+ 8 years)  treated in the same period (group II). Baseline clinical and echocardiographic characteristics were similar in the two groups. Four patients died during hospitalization, one aged > 75 years. In survived  patients at the end of follow up period sinus rhythm was present respectively in 64% of elderly and  74 % of younger patients. The rate of persistence of sinus rhythm without AF recurrences (38% vs.  41%) was similar in the two groups. After surgery sinus rhythm was more frequently never regained  in aged patients (27% vs. 20 %). Elderly patients + had an higher need for permanent pacing, more  frequent hospitalizations and an higher number of non AF atrial tachyarrythmias. At eight year  follow-up survival was lower in older patients (48 % aged >75 vs. 79% aged <75 years). The authors concluded that in elderly patients had a long term similar rate of stable sinus rhythm maintenance in comparison to younger patients after AF radiofrequency ablation associated with mitral valve surgery. However they needed more frequent permanent pacing, had higher rate of hospitalizations and post-proce- dural atrial tachyarrhythmias. The effects of survival are difficult to evaluate due to different life  expectancy of the two groups. 

Author Response

We thank you for your review .